# Offline Reinforcement Learning at Multiple Frequencies

**Kaylee Burns[1], Tianhe Yu[1,2], Chelsea Finn[1,2], Karol Hausman[1,2]**
[1]Stanford University  [2]Google Research
kayburns@cs.stanford.edu
Project Website: https://sites.google.com/stanford.edu/adaptive-nstep-returns/

**Abstract:** Leveraging many sources of offline robot data requires grappling with the heterogeneity of such data. In this paper, we focus on one particular aspect of heterogeneity: learning from offline data collected at different control frequencies. Across labs, the discretization of controllers, sampling rates of sensors, and demands of a task of interest may differ, giving rise to a mixture of frequencies in an aggregated dataset. We study how well offline reinforcement learning (RL) algorithms can accommodate data with a mixture of frequencies during training. We observe that the $Q$-value propagates at different rates for different discretizations, leading to a number of learning challenges for off-the-shelf offline RL. We present a simple yet effective solution that enforces consistency in the rate of $Q$-value updates to stabilize learning. By scaling the value of $N$ in $N$-step returns with the discretization size, we effectively balance $Q$-value propagation, leading to more stable convergence. On three simulated robotic control problems, we empirically find that this simple approach outperforms naïve mixing by 50% on average.

**Keywords:** offline reinforcement learning, robotics

## 1 Introduction

Given the cost of robotic data collection, we would like robots to learn from all possible sources of data. However, the robot data that is available may be heterogeneous, which can present challenges to offline reinforcement learning (RL) algorithms. One particular form of variability that we focus on in this work is the control frequency. Data collected independently may have been collected at different frequencies due to different sensor sampling rates or how the data was collected. For example, data col-

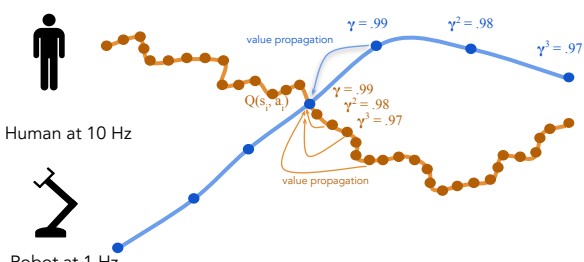

Figure 1: Training offline RL with diverse frequencies is challenging because the discrete MDP depends on the size of the timestep in between observations. This discretization affects the rate of value propagation, with smaller discretizations requiring more training updates to propagate the values.

lected through teleoperation may have a higher frequency to accommodate a more reactive and intuitive user interface, while data collected via online RL may use lower frequencies for more stable learning [1]. In light of the benefits from learning from large and diverse datasets [2, 3], our goal in this paper is to study whether offline RL algorithms can effectively learn from data with multiple underlying frequencies, and improve over learning separate policies for different frequencies.

While prior work studies how to stabilize RL algorithms at a single high frequency [1], effectively learning from data with multiple frequencies has been paid less attention. Work on time-series analysis has studied how to handle irregularly sampled data [4, 5], but not in the context of the approximate dynamic programming updates that arise in value-based reinforcement learning settings. As we will find in Section 4, running offline RL algorithms on data with multiple control frequencies leads to differing $Q$-values and diminished performance, a problem that, to our knowledge, has not been analyzed or addressed in previous research.

6th Conference on Robot Learning (CoRL 2022), Auckland, New Zealand.

A key insight in this work is that $Q$-learning from different data discretizations leads to different rates of value propagation: value propagates more quickly for coarser time discretizations. These different rates creates inconsistent regression targets, which can lead to high variance updates and training instability during optimization. Fortunately, we can use knowledge of the time discretization to normalize the rate of value propagation across different discretizations: our updates can look ahead to farther $Q$-values for data with a smaller discretization. This intuition leads us to a simple adjustment to $N$-step returns [6, 7] to align the discretizations, where we adaptively select the value of $N$ based on the time discretization of the particular trajectory.

The main contribution of this paper is to analyze and provide a solution to the problem of offline reinforcement learning from data with multiple time discretizations. First, we show that naïvely mixing data of different discretizations leads to diminished performance, and posit that this issue is caused by varying rates of value propagation for different discretizations. Then, we provide a simple technique that addresses this challenge via Adaptive $N$-Step returns. On three simulated robotic control problems, including two manipulation problems on simulated Sawyer and Panda robot arms, we find that our Adaptive $N$-Step update rule leads to stabler training and significantly improved performance, compared to naïve mixing or only using learning policies individually for each discretization.

## 2 Related Work

**Supervised learning of irregular time series.** Datasets with varying frequencies can arise whenever observations are pulled inconsistently from a continuous time system. Some approaches focus on consuming irregularly sampled data [8, 4, 5] by modeling the latent state of the data-consuming process as a differential equation. Unlike past work that integrates continuous time models into control tasks [9, 10], we are focused on the setting where the discretization remains constant within each trajectory, but varies within batch updates.

**Continuous time RL.** We focus on control tasks learned with offline RL, so in addition to learning a policy that can operate at multiple time scales, we also need to learn a value function at multiple time scales. Past work [11, 12, 13] uses the Hamilton-Jacobi Bellman equation to derive algorithms for estimating the value function. These formalisms have been used to create algorithms that can handle settings where the environment evolves concurrently with planning [14] and to learn to budget coarse and fine-grained time scales during training [15]. In addition to modeling challenges, continuous time introduces challenges during training as well. Tallec et al. [1] shows that deep $Q$-learning can fail in near-continuous time (i.e., fine-grained) settings and uses the formalisms from continuous RL to make $Q$-learning more reliable for small discretizations by using an advantage update.

**Offline RL.** Offline RL [16, 17, 18, 19] has emerged as a promising direction in robotic learning with the goal of learning a policy from a static dataset without interacting with the environment [20, 21, 22, 23]. Prior offline RL approaches focus on mitigating the distributional shift between the learned policy and the data-collecting policy [24] via either explicit or implicit policy regularization [25, 26, 27, 28, 29, 30, 31], penalizing value backup errors [32], uncertainty quantification [26, 33, 34], and model-based methods [35, 36, 37, 38, 39]. None of these works consider the practical problem of learning from offline data with different control frequencies, which is the focus of this work.

## 3 Preliminaries

A discrete-time MDP is a process that approximates an environment that is fundamentally continuous. For example, a robot may be moving continuously but receiving observations and sending actions at a fixed sampling rate, $1/\delta t$, where $\delta t$ is the time between the observations. A discrete-time MDP that takes into account $\delta t$ can be derived by discretizing a continuous-time MDP [12, 1].

Given a time-discretization $\delta t$, we arrive at the discrete MDP [1] $M_{\delta t} = (\mathcal{S}, \mathcal{A}, T_{\delta t}, r, \gamma)$, where $T_{\delta t}(s'|s, a)$ denotes the transition function over the timestep $\delta t$, $\mathcal{S}$ and $\mathcal{A}$ denote the state and action spaces, $r$ the reward function, and $\gamma \in (0, 1)$ the discount factor. The cumulative discounted return of this MDP can be written as: $R_{\delta t} := \sum_{k=0}^{\infty} \gamma^{k\delta t} r(s_k, a_k)\delta t$; and the $\delta t$-dependent $Q$-function can be written as: $Q_{\delta t}^{\pi} = r(s, a)\delta t + \gamma^{\delta t}\mathbb{E}_{\tau \sim \pi, T_{\delta t}}\left[\sum_{k=0}^{\infty} \gamma^{k\delta t} r(s_k, a_k)\delta t | s_0 = s\right]$. We will use the $\delta t$-dependent rewards and $Q$-functions to analyze the case of mixing data with different discretizations, i.e. with different values of $\delta t$, in offline RL. These definitions are similar to those of a standard Discrete-MDP, but with appropriate scalings for the given discretization, $\delta t$.

**Offline RL with conservative $Q$-learning.** Offline RL tackles the problem of learning control policies from offline datasets where the main challenge is the distribution shift between the learned policy and the behavior policy that was used to collect the data. A common offline RL approach that mitigates this issue focuses on an additional pessimism loss that encourages the $Q$-value update to stay close to the actions that it has seen in the data. One popular instantiation of this idea is conservative $Q$-learning (CQL) [32], which optimizes the following objective: $Q^{k+1} = \arg\min_Q \big[\alpha \cdot \big(\mathbb{E}_{s\sim\mathcal{D},a\sim\mu(a|s)}[Q(s,a)] - \mathbb{E}_{s\sim\mathcal{D},a\sim\hat{\pi}(a|s)}[Q(s,a)]\big) + \frac{1}{2}\mathbb{E}_{s,a,s'\sim\mathcal{D}}\big[\big(Q(s,a) - r(s,a) - \gamma\mathbb{E}_\pi(a|s)[Q^k(s,a)]\big)\big]\big]$, where $\mathcal{D}$ is the offline dataset of states, actions, and rewards, $\mu$ is a wide action distribution close to the uniform distribution, and $\hat{\pi}$ is the behavior policy that collects the offline data. We will show how we can use a modified version of CQL to incorporate data from different discretizations in offline RL.

## 4 Mixed Discretizations Can Destabilize Offline RL

In this section, we introduce the problem of simultaneously learning from several data sources with different frequencies. We show how differing rates of value propagation can create instability during training, which causes naively mixing frequencies to fail, and verify this intuition on a simple task.

**Problem setup.** Our objective is to learn a policy or set of policies that maximizes the expected return over a dataset that contains a mixture of different observation frequencies. Concretely, our goal is to learn $\pi^*(a|s,\delta t)$ over a distribution, $\Delta$, of discretizations: $\pi^*_{\delta t}(a|s) = \pi^*(a|s,\delta t) = \arg\max_\pi \mathbb{E}_{\delta t\sim\Delta}\left[\mathbb{E}_{\pi,T_{\delta t}}[R_{\delta t}]\right]$. To accomplish this goal, we utilize offline RL and focus our analysis on the behavior of $Q$-values in the presence of mixed discretizations. Specifically, we are interested in studying the objective $\mathbb{E}_{\delta t\sim\Delta}[\mathcal{L}_Q]$, where $\mathcal{L}_Q$ is any loss based on the Bellman equation.

We use CQL as our offline RL algorithm of choice, but our analysis is applicable to other value-based offline RL methods. We modify the CQL objective described in Sec. 3 to incorporate a distribution of discretizations, $\delta t$ as follows:

$$Q_{\delta t}^{k+1} = \arg\min_Q \mathbb{E}_{\delta t\sim\Delta}\left[\alpha \cdot \big(\mathbb{E}_{s\sim\mathcal{D}_{\delta t},a\sim\mu(a|s)}[Q_{\delta t}(s,a)] - \mathbb{E}_{s\sim\mathcal{D}_{\delta t},a\sim\hat{\pi}_{\delta t}(a|s)}[Q_{\delta t}(s,a)]\big)\right.$$
$$\left. + \frac{1}{2}\mathbb{E}_{s,a,s'\sim\mathcal{D}_{\delta t}}\left[\big(Q_{\delta t}(s,a) - r(s,a)\delta t - \gamma^{\delta t}\mathbb{E}_{\pi_{\delta t}(a'|s')}[Q_{\delta t}^k(s',a')]\big)^2\right]\right]$$

The $Q$-targets of different $\delta t$ may receive value updates at different rates because it takes longer for value to propagate when the state space is divided by more fine-grained steps. We study the challenges this creates for $Q$-learning next. We refer to this unmodified objective as Naïve Mixing.

**Value propagates in algorithmic time, not physical time.** A key challenge in mixing data of different discretizations together is that algorithmic time, $k$, scales inversely with the size of the discretization: $k = \frac{t}{\delta t}$. This implies that value propagates along the state space at a rate that is proportional to the discretization. This phenomenon is illustrated for a simple gridworld environment in Figure 2. The agent starts on the right side and receives a reward of 10 for moving forward to the left. In the top row, the agent can take steps of size 1 and in the bottom row the agent can take steps of up to size 2, which simulates policies that operate at different frequencies. At every step of the algorithm, $k$, we perform the update rule $V_k^{\delta t}(s) = \max_a r(s,a) + \gamma V_{k-1}^{\delta t}(s')$. Although in this toy example, we are performing tabular updates, we can see how the nature of the Bellman update generates a distribution of targets that is difficult for a function approximator, like a neural network, to match. For example, at step $k = 3$, a network would receive the following targets for the first three states from the start: 0 for $\delta t = 1$ and 5.3 for $\delta t = 2$ for the first and second state, and 0 for $\delta t = 1$ and 6.6 for $\delta t = 2$ for the third state. We posit that the rate at which the targets are updated for different discretizations and the resulting difference in the target values leads to training instability and subpar performance. In the next section, we verify this intuition empirically.

### 4.1 Analyzing Naïve Mixing

We study how offline RL behaves when naïvely mixing discretizations in a pendulum environment and posit that differing rates of $Q$-function updates lead to training instability that hurts performance.

**Setup.** To test our hypothesis, we use the previously-used simple pendulum environment from the OpenAI Gym [1]. We modify the pendulum environment by changing the underlying time

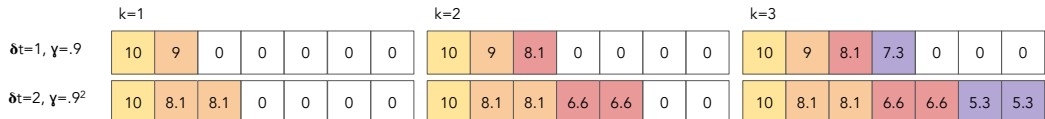

Figure 2: We visualize how value propagates for different $\delta t$ in a simple gridworld. In the top row, the agent takes actions of length 1 and in the bottom of maximum length 2. A reward of 10 is attained at the goal state in the left-most square of the grid. For each update step $k$, the value propagates twice as much in the second setting, creating regression targets that are difficult to match. For example, at step $k = 2$, the $Q$-value needs to regress to targets 6.6 and 0 from the same states. Boxes are color-coded by update step.

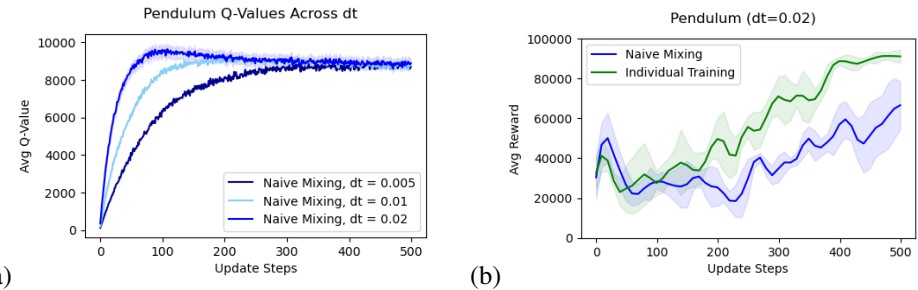

(a)                                                                                              (b)

Figure 3: We analyze training with mixed discretizations for the pendulum swing up task. (a) Over the course of training, the average $Q$-value scales with $\delta t$. As a result, the $Q$-value of more fine-grained discretizations propagates more slowly than for coarser discretizations. (b) Compared to training individually, the final performance of a policy trained for $\delta t = 0.02$ is corroded by adding more data of different discretizations.

discretization of the simulator and collect data with discretizations $\delta t \in \{0.02, 0.01, 0.005\}$ seconds. We make the reward function for this task sparse by creating a window of angles and velocities around 0 where the algorithm receives a constant reward. This increases the difficulty of the task and gives us a more precise understanding of the comparisons between the $Q$-values. Without this modification naive mixing converges in under 10 gradient steps. We first collect data by running Deep Advantage Updating (DAU) [1] for each discretization individually and then store buffers of $500k$ observations per discretization. The data collected is comparable to an expert-replay dataset. We combine the data coming from the three different discretizations together into a single offline dataset.

**Value propagation.** We look at the average $Q$-value for different discretizations over the course of training. The average cumulative reward in the dataset for each $\delta t$ is similar, so we expect the average $Q$-value of each discretization to be the same. However, because more fine-grained data propagates the sparse reward at a slower rate than for coarser data as shown in the previous subsection, we see in Figure 3 (a) that the coarser discretization of $\delta t = 0.02$ reaches a higher average $Q$-value more quickly than a fine-grained discratization of $\delta t = 0.01$, which in turn reaches a higher $Q$-value more quickly than the most fine-grained discretization $\delta t = 0.005$. By the end of training the average $Q$-values across different $\delta t$ converge. This confirms the hypothesis that the $Q$-values grow at different rates because of the nature of the Bellman update and not because the true value is different.

**Optimization challenges.** Next, we want to understand if the different rates of the $Q$-value update for different discretizations cause optimization challenges during offline RL. In Figure 3 (b) we plot performance over the course of training with and without adding data. Ideally our algorithms would be aided by the addition of more data, but after introducing data of different frequencies, we see the average return across training drop for $\delta t = 2$.

To summarize, these results suggest that, even within a simple control task, naïvely mixing data with different discretizations hinders the performance by slowing value propagation and introducing optimization challenges. Next, we present a method that aims to avoid these issues so that additional data can be leveraged effectively.

## 5  Adaptive $N$-Step Returns

The goal of our method is to "align" the Bellman update rate for different discretizations so that they can be updated at a similar pace, creating more consistent value targets. Intuitively, we aim to mimic

**k=1**

| δt=1, γ=.9 | 10 | 9 | 8.1 | 0 | 0 | 0 | 0 |
|---|---|---|---|---|---|---|---|
| δt=2, γ=.9² | 10 | 8.1 | 8.1 | 0 | 0 | 0 | 0 |

**k=2**

| δt=1, γ=.9 | 10 | 9 | 8.1 | 7.3 | 6.6 | 0 | 0 |
|---|---|---|---|---|---|---|---|
| δt=2, γ=.9² | 10 | 8.1 | 8.1 | 6.6 | 6.6 | 0 | 0 |

**k=3**

| δt=1, γ=.9 | 10 | 9 | 8.1 | 7.3 | 6.6 | 5.9 | 5.3 |
|---|---|---|---|---|---|---|---|
| δt=2, γ=.9² | 10 | 8.1 | 8.1 | 6.6 | 6.6 | 5.3 | 5.3 |

Figure 4: We modify value iteration on a simple gridworld environment. In the bottom row, the agent can take actions that are double the size of the other agents actions. However, because the agent in the top row uses an $N$-step update rule with $N = 2$, the value propagates along the state space at the same rate for each agent.

the "update stride" of coarser discretizations when training on more fine-grained discretizations to bring consistency to the $Q$-value updates.

The core idea behind our approach is to utilize $N$-step returns as a tool that accelerates the propagation of $Q$-values. In particular, we calculate $Q$-targets with Adaptive $N$-Step returns, where we scale $N$ by the value of $\delta t$. To better understand the idea behind this approach, we revisit the same intuition as presented in Figure 2, but with the updated $N$-step modifications in Figure 4. In this case, we use the following value iteration update rule: $V_k(s_t) = \sum_{t'=0}^{\frac{N}{\delta t}-1} \gamma^{t'} r(s_{t+t'}, a_{t+t'}) + \gamma^{\frac{N}{\delta t}} V_{k-1}(s_{t+\frac{N}{\delta t}})$. This update rule is equivalent to $N$-step returns for value iteration, but where the number of backup steps, $N$, is scaled by $\delta t$. Figure 4 shows an example with $N = 2$, where the top row uses $\delta t = 1$ and the bottom row uses $\delta t = 2$. When we adjust the number of backup steps by the frequency $\delta t$, to $\frac{N}{\delta t} = \frac{2}{1} = 2$ and $\frac{N}{\delta t} = \frac{2}{2} = 1$ in the top and bottom respectively, we see that the differences in the targets available to a function approximator are significantly smaller in magnitude than in Figure 2. Note that the slight differences that remain are due to the precision allowed by the discretization.

Our normalization scheme can be implemented as a Q-value update rule that can be integrated into any offline RL algorithm that learns $Q$-values. At every update step, we calculate the target $Q$ at $\delta t$ as:

$$Q_{\text{target}}^{\delta t} = \sum_{t'=0}^{\frac{N}{\delta t}-1} \gamma^{t'} r(s_{t+t'}, a_{t+t'}) + \gamma^{\frac{N}{\delta t}} Q^{\delta t}(s_{t+\frac{N}{\delta t}}, a_{t+\frac{N}{\delta t}}) \tag{1}$$

In all of our experiments, we select $N$ to be the largest $\delta t$ present in the dataset (i.e., the most coarse discretization). With this choice of $N$, the more fine-grained discretizations exactly match the update stride of the coarsest discretization under standard offline RL.

This update rule applies to any Q-learning algorithm. In particular, we use CQL in our experiments. This requires adapting both the standard Bellman loss as well as the pessimism and optimism terms, which are $\mathbb{E}_{s\sim\mathcal{D}, a\sim\mu(a|s)}[Q^{\delta t}(s,a)]$ and $\mathbb{E}_{s\sim\mathcal{D}, a\sim\hat{\pi}(a|s)}[-Q^{\delta t}(s,a)]$ respectively. We replace the Bellman operator in Equation 1 with $N$-step returns scaled by discretization. The pessimism and optimism terms are applied to the state and action at $\frac{N}{\delta t}$, where the target $Q$-function is evaluated. This results in the complete objective:

$$\mathcal{L}_{\text{CQL}} = \max_{\mu} \alpha \mathbb{E}_{s\sim\mathcal{D}_{\delta t}}[\mathbb{E}_{a\sim\mu(a|s)}[Q^{\delta t}(s_{t+\frac{N}{\delta t}}, a_{t+\frac{N}{\delta t}})] - \mathbb{E}_{a\sim\hat{\pi}(a|s,\delta t)}[Q^{\delta t}(s_{t+\frac{N}{\delta t}}, a_{t+\frac{N}{\delta t}})]] + \mathcal{R}(\mu)$$

$$\mathcal{L}_{\text{N-Step}} = \mathbb{E}_{\delta t\sim\Delta}\left[\mathcal{L}_{\text{CQL}} + \frac{1}{2}\mathbb{E}_{s_t, a_t, s_{t+1}\sim\mathcal{D}_{\delta t}}\left[\left(Q^{\delta t}(s_t, a_t) - Q_{\text{target}}^{\delta t}\right)^2\right]\right]$$

If $\frac{N}{\delta t}$ is not an integer, we sample the number of steps from the nearest digits (i.e., rounding up or down) with a Bernoulli distribution where the success rate parameter $p$ is the fractional part of $\frac{N}{\delta t}$.

## 6 Experiments

In Section 4, we showed how naively mixing data with different discretizations updates the $Q$-function at different rates leading to instability in training. In our experiments, we test this trend on popular benchmarks and validate that our method proposed in Section 4 can remedy these challenges. In particular, we seek to answer the following questions: (1) Does Adaptive $N$-Step returns fix the performance challenges introduced by mixing different discretizations? (2) What impact does Adaptive $N$-Step returns have on the learned $Q$-value during training? (3) What is the influence of Adaptive $N$-Step on training stability? (4) How does leveraging multiple discretizaiton data sources compare to learning only from a single discretization? (5) How much of the improvement can be attributed to general benefits from $N$-step returns compared to the synchronization of $N$?

| Env Name | $\delta t$ | Naïve mixing | Adaptive N-step (ours) |
|---|---|---|---|
| pendulum | .02 | $64.8 \pm 8.4$ | $\mathbf{91.1} \pm 3.7$ |
| | .01 | $57.5 \pm 10.5$ | $\mathbf{81.5} \pm 2.4$ |
| | .005 | $37.5 \pm 3.5$ | $\mathbf{62.0} \pm 6.3$ |
| | Avg | $53.3 \pm 7.5$ | $\mathbf{78.2} \pm 4.2$ |
| door-open (max success) | 10 | $\mathbf{100.0} \pm 0.0$ | $89.5 \pm 8.1$ |
| | 5 | $\mathbf{95.0} \pm 3.9$ | $92.9 \pm 5.7$ |
| | 2 | $\mathbf{100.0} \pm 0.0$ | $\mathbf{100.0} \pm 0.0$ |
| | 1 | $5.7 \pm 4.2$ | $\mathbf{87.5} \pm 7.5$ |
| | Avg | $75.2 \pm 2.0$ | $\mathbf{92.5} \pm 5.3$ |
| kitchen-complete-v0 | 40 | $20.2 \pm 5.7$ | $\mathbf{34.6} \pm 8.7$ |
| | 30 | $9.3 \pm 4.4$ | $\mathbf{19.9} \pm 7.2$ |
| | Avg | $14.7 \pm 5.0$ | $\mathbf{27.3} \pm 7.9$ |

Table 1: Performance of Naïve Mixing and Adaptive $N$-Step. Within the sparse reward environments of pendulum and kitchen, Adaptive $N$-Step outperforms Naïve Mixing when evaluated on any $\delta t$. For the dense reward door-open task, Adaptive $N$-Step enables strong performance on $\delta t = 1$ while maintaining strong performance on coarser discretizations.

## 6.1 Experimental Setup

**Tasks and datasets.** We evaluate Adaptive $N$-Step on three different simulation environments of varying degrees of difficulty described below.

*Pendulum.* Similarly to Section 4, we used a modified Open AI Gym [40] pendulum environment. We modify it by changing the underlying discretization of the simulator and collect data with disctretizations of $0.02, 0.01, 0.005$ by training DAU [1] on each discretization independently and storing the replay buffer, which contains $500k$ observations. We use a sparse reward of 100 scaled by $\delta t$ that is attained in a range of angles and velocities around 0. In all mixing baselines, we condition the $Q$-value and policy on the discretization to train a single RL algorithm on all discretizations.

*Meta-World.* We modify the door environment from Meta-World [41] to support different discretizations by adjusting the frame-skip. We experiment with frame skips of $1, 2, 5$, and $10$. We focus on door-open because it is a straightforward manipulation task. In this environment, we find not conditioning on $\delta t$ to give stronger performance for our method as well as the baselines. To evaluate performance, we look at the maximum success score, which measures whether or not the environment was in the goal configuration at any point during the trajectory.

*FrankaKitchen.* FrankaKitchen [42] is a kitchen environment with a 9-DoF Franka robot that can interact with 5 household objects. We study the long-horizon task sequence of opening the microwave, moving the kettle, turning on a light, and opening a drawer with a sparse reward. Upon completion of any of the four tasks, the agent receives a reward of 1. We mix data with frame skips of 30 and 40. The data at 40, which is the default frame skip, comes from the D4RL dataset [43]. Specifically, we use the expert data called `kitchen-complete-v0`. To generate data at a frame skip of 30, we run a trained policy at that frame skip. The quality of this policy is lower, making the data at 30 most similar to a medium-replay dataset.

**Implementation and training details.** We build on top of the CQL implementation from Geng [44], and use the following CQL hyperparameters: all of our models use a CQL alpha term of 5, a policy learning rate of $3e-5$, and a $Q$-function learning rate of $3e-4$. For Pendulum we use a discount scaled with the size of $\delta t$, specifically $.99^{\frac{\delta t}{\max_{\delta t \in \Delta} \delta t}}$. For Meta-World, we found a constant discount factor of $.99$ to work best across the experiments. Each task has a maximum trajectory length of 500 regardless of the discretization size. We implement $Q^{\delta t}$ as an optionally $\delta t$-conditioned neural network by adding a normalized $\delta t$ feature to the state.

## 6.2 Results

**Final performance with and without $N$-step.** To answer question (1) (*Does $\delta t$-scaled $N$-step returns fix the performance challenges introduced by mixing different discretizations?*), we compare the normalized performance at the end of training with and without N-step modification in

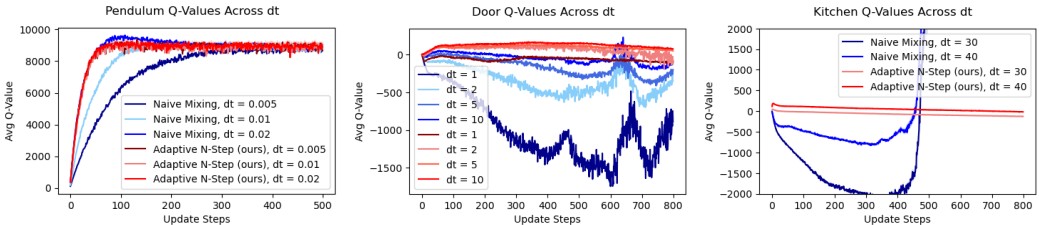

Figure 5: Adaptive $N$-step results in more consistent q-values across discretizations and tasks. In all figures, blue denotes naive mixing and red denotes Adaptive $N$-step returns.

Table 1. Results are averaged and reported with 95% confidence intervals over runs from 5 seeds with 5 evaluations from each run. For the sparse reward tasks – pendulum and kitchen – $\delta t$ scaled $N$-step significantly improves final training performance across discretizations, resulting in an almost double average return in the case of kitchen, and more than $50\%$ improvement in pendulum. In the dense reward tasks, i.e. door-open, $N$-step provides an advantage for more fine-grained discretizations without compromising the high performance naïve mixing already achieves on larger discretizations. These results confirm that our method improves mixing discretization performance across the benchmark tasks. Next, we aim to understand its impact on important quantities such as $Q$-values.

**Q-values across training.** In Section 4, we observed that the $Q$-value learned with the naïve mixing strategy is updated differently for different discretizations. We aim to confirm this hypothesis on other, more complex environments as well as analyze whether our adaptive $N$-step method can remedy this issue and lead to more consistent $Q$-value updates. To accomplish this goal, we present Fig. 5, where we visualize the average $Q$-value for all discretizations with and without adaptive $N$-step method. When we compare the $Q$-values after using adaptive $N$-step, we observe that the $Q$-values become more consistent across discretizations. In the top left of Figure 5, we see that the $Q$-values obtained for pendulum are nearly identical for different discretizations across the course of training, resolving the issue pointed out in Sec. 4. Interestingly, we also note that the $Q$-values of naïve mixing eventually converge to the same value as $N$-step, but the performance of the policy learned with these values is still diminished as previously described in Table 1. This suggests that the consistency of the updates across training and not just the absolute value of the $Q$-function are important in learning a good policy. In the top right and bottom left plots in Fig. 5, we see that, in Meta-World, Adaptive $N$-Step returns has two effects. It makes the average $Q$ higher for all discretizations and minimizes the difference across discretizations similarly to the pendulum experiment. Within Kitchen, $N$-step has the same effect of pulling up the $Q$-values and has the added effect of minimizing the standard error of the $Q$-value across seeds. With naïve mixing, on the other hand, the $Q$-values are inconsistent throughout training and, contrary to pendulum, they end up exploding by the end of training. These experiments indicate a positive, stabilizing impact of Adaptive $N$-Step on the learned Q-values during training, answering question (2).

**Performance across training.** To answer question (3) (*What is the influence of our method on training stability?*), we plot the evaluation performance over the course of training averaged over all discretizations in each environment. The results for Pendulum, Meta-World Door, and Kitchen tasks are visualized in Figure 6. For Kitchen and Pendulum, reward corresponds to task success so we plot the average reward. For the Door task in Meta-World, to better reflect the task, we plot the maximum success attained in the evaluation trajectory. We observe a significant difference in performance and training stability in Pendulum and Kitchen when comparing Adaptive $N$-Step and naïve mixing. In particular, the Kitchen task indicates that naïve mixing experiences substantial instability in training, where around 200 steps into the training the performance goes down. This showcases that the performance is unstable even when a reasonable policy can be learned at some point during training. The performance plot for the Door task shows comparable performance when all discretizations are averaged together, however, we refer back to Table 1 to show that certain discretizations (i.e. $\delta_t = 1$) perform significantly worse than others compared to the Adaptive $N$-Step strategy.

**Comparison to individual training.** Following our motivation for this work, we investigate question (4), whether mixing multiple discretizations with our method outperforms training on a single discretization. In mixing diverse data sources together we hope to leverage the data from each source

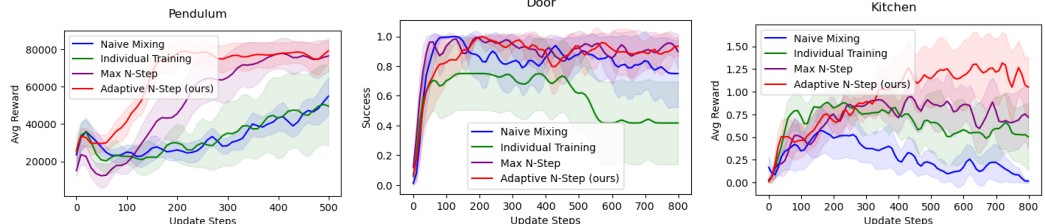

Figure 6: Average performance over the course of training. Without Adaptive $N$-Step returns, naïve mixing can suffer from instability. Even though Max $N$-Step uses a larger value of $N$ for most discretizations, Adaptive $N$-Step still converges to a high average reward more quickly than Max $N$ on Pendulum and attains a higher final average performance on Kitchen.

to learn a model that performs better than a set of independently trained models. We train separate policies on the dataset for each discretization and present the averaged performance in Fig. 6 as Individual Training. The results clearly indicate that training individually on a single discretization underperforms relative to training with Adaptive $N$-Step returns across discretizations.

**Comparison to max-N step.** We study question (5) to understand the importance of synchronization of the update stride of the $Q$-values. Another explanation of the performance gain from Adaptive $N$-Step is simply that $N$-Step returns is a good design choice for reducing variance across discretizations. To explore this possibility, we create an additional baseline, Max $N$-Step Returns, where we use the largest adapted $N$ for all discretizations present in our dataset. We modify the $Q$-target calculation as follows:

$$N_{\max} = \frac{N}{\min_{\delta t \in \Delta} \delta t}$$

$$Q_{\text{target}}^{\delta t} = \sum_{t'=0}^{N_{\max}-1} \gamma^{t'} r(s_{t+t'}, a_{t+t'}) + \gamma^{N_{\max}} Q^{\delta t}(s_{N_{\max}}, a_{N_{\max}})$$

In Fig. 6, we see that Adaptive $N$-Step returns converges to a high average reward more quickly than Max $N$-Step in the Pendulum environment. This is surprising because Max $N$ uses a larger value of $N$ for $\delta t = .01$ and $\delta 1 = .005$ and therefore could propagate reward more quickly than Adaptive $N$. In the Door Open task, Max $N$ performs similarly to Adaptive $N$ across discretizations. For FrankaKitchen, Adaptive $N$-Step returns exceeds the final average performance of Max $N$-Step returns.

## 7 Limitations

Although our adaptive $N$-step method works reliably for discretizations where the step sizes can be aligned, it is unclear how well adaptive $N$ could work when the frequencies do not have a small common multiple. In the kitchen setting, we saw that the frequencies do not need to be multiples of one another to reap the benefits of adaptive $N$-step returns, but this strategy could become less viable as the least common multiple between frequencies grows. Other strategies, such as sampling different $N$ with probability proportional to the fractional difference of the two frequencies may need to be adopted. Another limitation of this work is that the discretization of each trajectory is assumed to be fixed and given. If the frequency of observations varies within a trajectory, the value of $N$ may need to be predicted from observations.

## 8 Conclusion

In this work, we studied the problem of mixing data from different discretizations in offline RL and presented a simple and effective approach for this problem that improves final performance across three simulated control tasks. We found that our Adaptive $N$-Step returns method counteracts varying rates of value propagation present when mixing data naïvely. While our approach provides a step towards offline RL methods that can consume data coming from diverse sources, more future work is needed to address training offline RL agents on other sources of variation that are commonly present in robotics data.

**Acknowledgments**

We thank Google for their support and anonymous reviewers for their feedback. We're grateful to Evan Liu and Kyle Hsu for reviewing drafts of our submission and Annie Xie for providing references to implementations. Kaylee Burns is supported by an NSF fellowship. Chelsea Finn is a CIFAR fellow. This work was supported by ONR grant N00014-22-1-2621.

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
