# OpenReview forum: "Offline Reinforcement Learning at Multiple Frequencies"
_robot-learning.org/CoRL/2022/Conference — CoRL 2022 Poster_

### Official Review · Reviewer_W1tF · 2022-07-21

**Originality:** Good
**Technical Quality:** Good
**Clarity Of Presentation:** Very Good
**Impact:** 3

**Recommendation:**

Weak Accept: I recommend accepting the paper, but will not argue for my recommendation if the majority of other reviewers have a different opinion.

**Summary:**

The paper considers a novel setting of offline reinforcement learning from datasets collected at different control frequencies. This setting presents challenges for approximate dynamic programming since the value functions are different for each control frequency. A naïve mixing of datasets with different control frequencies is shown to cause unstable training in the pendulum control task. The key idea in the paper is to use adaptive N-step value targets where N is determined based on the control frequency of the sample. This is shown to stabilize training in three different tasks (pendulum, door, and kitchen), making effective use of the heterogeneous datasets of different control frequencies.

**Issues:**

See weaknesses listed above.

* Experiments on tasks with different optimal values for policies with different control frequencies would provide more intuition about the proposed approach.

* Tasks with very small $\delta t$ are known to cause problems for RL [1] and using DAU instead of SAC as the basis of CQL could help improve the performance in these settings.

* Results on more tasks such as the D4RL benchmark would further support the contributions of the paper.

* Figure 6.2 should be Figure 7 in L283.

**Quality Of The Limitations Section:**

Limitations are addressed clearly

**Reviewer Expertise:**

4: The reviewer is confident but not absolutely certain that the evaluation is correct

**Robotics Focus:**

Highly relevant to robotics but no hardware experiments

**Strengths And Weaknesses:**

Strengths:

* The paper explores a novel setting of offline RL from trajectories collected at different control frequencies. This is a realistic setting in robot learning and an effective algorithm could improve the possibilities of offline RL.

* The proposed approach is simple to implement and understand. It significantly outperforms the naïve mixing strategy in the tested benchmark tasks.


Weaknesses:

* I believe the key problem in the considered setting of RL at different frequencies is that the optimal value function is different for policies acting at different frequencies. For example, a policy acting at a higher frequency could perform much better in a task that requires such quickly changing actions. It is not clear how the proposed approach solves this problem.

* The proposed solution consists of the adaptive N-step returns and also the conditioning of the actor-critic networks on the $\delta t$ value. The influence of the latter on the agent performance is not clear. If the problem in the offline RL at multiple frequencies setting is that value propagation occurs at different rates, isn’t it theoretically possible that this doesn’t cause any training instabilities since the critic network could predict the right value target given a particular control frequency? For example, in Figure 2, it is argued that the Q network needs to regress to different targets from the same states but if the Q network is conditioned on $\delta t$ then it is correctly trained to predict different targets based on the control frequency. An ablation study with naïve mixing and $\delta t$-conditioned actor-critic networks would be interesting.



**Summary Of Recommendation:**

The problem setting considered in the paper is novel and the proposed solution is shown to be effective. However, the overall influence of the proposed adaptive N-steps is not clear, and more ablation studies have to be performed to better support the contributions of the paper.

---

> ### Author Response · Authors · 2022-08-23
> **Author Response to Reviewer W1tF**
>
> We greatly appreciate your insightful comments. We are updating the paper to address your points about different optimal policies at different frequencies and DAU as an additional baseline, which has improved the paper quality.
>
> > Tasks with very small $\delta t$ are known to cause problems for RL [1] and using DAU instead of SAC as the basis of CQL could help improve the performance in these settings.
>
> There are several components to DAU:
> 1. Scaling quantities like reward and discount by $\delta t$
> 2. Learning an advantage function, which, unlike the Q-function, has a limit as $\delta t \rightarrow 0$
> 3. Time-step invariant exploration
>
> Our naive mixing baseline includes all of the relevant scalings from (1). Component (3) is not relevant to our setting because we are learning from offline data. Below we show results from adding component (2) to the kitchen setting. We try both the default DAU implementation and also an offline modification of DAU where we add all of the additional CQL losses applied to the advantage function with a CQL alpha value of 5.
> | Env Name | $\delta t$ | Naive mixing with DAU scalings | DAU | DAU+CQL ($\alpha$=5) | Adaptive N-Step |
> | ----------- | ----------- | ----------- | ----------- | ----------- | ----------- |
> | FrankaKitchen | 40 | 20.2 $\pm$ 5.7 | 0.0 $\pm$ 0.0 | 0.0 $\pm$ 0.0 | **34.6** $\pm$ 8.7 |
> | | 30 | 9.3 $\pm$ 4.4 | 0.0 $\pm$ 0.0 | 0.0 $\pm$ 0.0 | **19.9** $\pm$ 7.2 |
> | | Avg | 14.7 $\pm$ 5.0 | 0.0 $\pm$ 0.0 | 0.0 $\pm$ 0.0 | **27.3** $\pm$ 7.9 |
>
> The DAU objective is not compatible with offline reinforcement learning, which likely explains its failure here. In the approximate advantage formulation of DAU, the value and advantage networks are updated with the following loss:
>
> $$Q^i \leftarrow V_{\theta}(s_i) +  \delta t(\bar{A}_\psi(s^i, a^i)-\max_a\bar{A}_\psi(s^i, a))$$
> $$\tilde{Q}^i \leftarrow r^i\delta t + (1-d^i) \gamma^{\delta t} V_\theta (s^{i+1}))$$
> $$ \mathcal{L} = \|| (Q^i-\tilde{Q}^i) \frac 1 {\delta t} ||_2^2 $$
>
> In practice $\tilde{Q}^i$ is a target network updated with Polyak averaging, so the effect of this loss in an offline setting is to minimize $\bar{A}_\psi(s^i, a^i)$ and maximize $\max_a\bar{A}_\psi(s^i, a)$. This directly counteracts the CQL objective, which enforces pessimism on actions that maximize q-values and optimism on actions within the replay buffer. We believe this conflict is why DAU fails. Further research is necessary to come up with a robust version of offline DAU. We’ll include these comparisons in the final version of the paper and discuss the applicability of DAU to the offline RL setting.
>
> > The proposed solution consists of the adaptive N-step returns and also the conditioning of the actor-critic networks on the δt value. The influence of the latter on the agent performance is not clear.
>
> We treat $\delta t$ conditioning as a hyperparameter that is set by tuning Naive Mixing (that is, we select whether or not to condition the $Q$-values and the policy on $\delta t$ by picking whichever achieves stronger performance on Naive Mixing). In Pendulum, we condition on $\delta t$ for both Naive Mixing and Adaptive $N$-Step returns, but in Meta-World and FrankaKitchen we don’t condition on $\delta t$. Below we re-run FrankaKitchen on 3 seeds conditioning both Naive Mixing and Adaptive $N$-Step on $\delta t$ and report standardized average return acros 5 runs with 95% confidence intervals:
>
> **FrankaKitchen**
> | $\delta t$ | Naive Mixing | Adaptive N-Step |
> | ----------- | ----------- | ----------- |
> | 40 | 1.7 $\pm$ 3.3 | **48.3** $\pm$ 3.3 |
> | 30 | 8.3 $\pm$ 16.3 | **16.6** $\pm$ 8.6 |
>
> Conditioning on $\delta t$ is still unstable for naive mixing, but boosts total average performance for Adaptive $N$-Step.
>
> > Experiments on tasks with different optimal values for policies with different control frequencies would provide more intuition about the proposed approach.
>
> Our method can learn different optimal values with $\delta t$ conditioning. To see this we can take the policies trained in the above experiment and condition the networks on the incorrect $\delta t$ feature and conduct the same evaluation:
>
> | Policy Conditioning | $\delta t$ | Adaptive N-Step |
> | ----------- | ----------- | ----------- |
> | $\pi_{\delta_t = 40}(a\|s)$ | 40 | 48.3 $\pm$ 3.3 |
> | $\pi_{\delta_t = 30}(a\|s)$ | 40 | 21.1 $\pm$ 6.5 |
> | $\pi_{\delta_t = 30}(a\|s)$ | 30 | 16.6 $\pm$ 8.6 |
> | $\pi_{\delta_t = 40}(a\|s)$ | 30 | 11.7 $\pm$ 14.2 |
>
> We observe a significant performance drop when using the incorrect $\delta t$, which suggests that the policy has learned different optimal values at different control frequencies
>
> > Results on more tasks such as the D4RL benchmark would further support the contributions of the paper.
>
> Currently, the FrankaKitchen data with a frameskip of 40 is from D4RL. We will make the dataset sources for each environment more clear in the final version of the paper.

---

> ### Author Response · Authors · 2022-08-25
> **Follow Up to Reviewer W1tF**
>
> Hi reviewer W1tF, We wanted to follow up to see if the response addresses your concerns or if you have any further questions. Thank you again!

---

### Official Review · Reviewer_Avb9 · 2022-07-27

**Originality:** Fair
**Technical Quality:** Good
**Clarity Of Presentation:** Very Good
**Impact:** 3

**Recommendation:**

Weak Accept: I recommend accepting the paper, but will not argue for my recommendation if the majority of other reviewers have a different opinion.

**Summary:**

The paper presents a method for offline reinforcement learning (RL) from heterogeneous datasets, where the trajectories are collected at different control frequencies. The paper identifies a key issue that the rate of value propagation via dynamic programming for data collected with different control frequencies can vary significantly, causing off-the-shelf offline RL algorithms to be unstable in practice. The authors propose a method to balance the rate of Q-value propagation by scaling the N-step returns by the discretization timestep. Experiments on three simulated control tasks demonstrate that this simple modification can stabilize offline reinforcement learning and lead to superior performance than naively using data from different discretizations or using only a subset of data with a fixed control frequency.


**Issues:**

I will consider changing the recommendation if the authors can address the following issues:

1. Provide results of exhaustive ablations over the time discretizations and empirical evidence of generalization to discretizations not seen during training.

2. Compare against stronger baselines such as [1].

3. Although this might be hard to achieve given the limited time, demonstration of the method on a real-world robotics task will be convincing for a strong accept.


**Quality Of The Limitations Section:**

Limitations are addressed clearly

**Reviewer Expertise:**

5: The reviewer is absolutely certain that the evaluation is correct and very familiar with the relevant literature

**Robotics Focus:**

Highly relevant to robotics but no hardware experiments

**Strengths And Weaknesses:**

**Strengths**

The paper is well motivated and the writing is clear and concise. The proposed algorithm is easy to understand, and the toy experiments on grid world and simple pendulum clearly illustrate the issues with Q-value propagation at different time frequencies. The experimental results demonstrate how the proposed method can make Q-function updates more consistent across different discretizations as well as lead to superior performance and stable learning as compared to naive mixing or using data only from a single discretization.

**Weaknesses**

The major concerns center around the lack of exhaustive ablations and strong baselines in the experimental section. The method is tested on a small set of hand selected discretizations and an exhaustive sweep over the control timesteps is missing. Since the method is only tested on discretizations that were present in the training data, it is unclear if the learned Q-function will generalize to time discretizations that were not seen during training, which is likely to occur in real-world robotics. The current experiments are limited to only 3 simulated tasks which also leaves the question unanswered if this would transfer well to real-world robots.
Furthermore, the baselines considered do not include other methods that attempt to make Q-learning robust to the time discretization [1]. Even in the current results, naive mixing outperforms the proposed method on one of the tasks, and without proper ablations, it is difficult to judge the true impact of adaptive N-step updates.

On a more technical level, the use of N-step returns would make the algorithm on-policy which could break Q-learning based algorithms. It would be helpful if the authors can comment on why this is not the case in their implementation.

[1] Tallec et. al; 2019, Making Deep Q-learning Methods Robust to Time Discretization



**Summary Of Recommendation:**

The reviewer’s recommendation for a weak reject is based on insufficient experimental evidence for the efficacy of the approach as well as unclear impact on real-world robotics. While the authors have identified an important sub-problem in offline reinforcement learning from heterogeneous datasets, in the current form the paper is not fit for publication.

---

> ### Author Response · Authors · 2022-08-23
> **Author Response to Reviewer Avb9**
>
> Thank you for your review. Following your suggestions to add an offline DAU baseline and interpolation and extrapolation results helped us better study the performance of Adaptive $N$-Steps.
>
> > Compare against stronger baselines such as [1].
>
> There are several components to DAU:
> 1. Scaling quantities like reward and discount by $\delta t$
> 2. Learning an advantage function, which, unlike the Q-function, has a limit as $\delta t \rightarrow 0$
> 3. Time-step invariant exploration
>
> Our naive mixing baseline includes all of the relevant scalings from (1). Component (3) is not relevant to our setting because we are learning from offline data. Below we show results from adding component (2) to the kitchen setting. We try both the default DAU implementation and also an offline modification of DAU where we add all of the additional CQL losses applied to the advantage function with a CQL alpha value of 5.
> | Env Name | $\delta t$ | Naive mixing with DAU scalings | DAU | DAU+CQL ($\alpha$=5) | Adaptive N-Step |
> | ----------- | ----------- | ----------- | ----------- | ----------- | ----------- |
> | FrankaKitchen | 40 | 20.2 $\pm$ 5.7 | 0.0 $\pm$ 0.0 | 0.0 $\pm$ 0.0 | **34.6** $\pm$ 8.7 |
> | | 30 | 9.3 $\pm$ 4.4 | 0.0 $\pm$ 0.0 | 0.0 $\pm$ 0.0 | **19.9** $\pm$ 7.2 |
> | | Avg | 14.7 $\pm$ 5.0 | 0.0 $\pm$ 0.0 | 0.0 $\pm$ 0.0 | **27.3** $\pm$ 7.9 |
>
> The DAU objective is not compatible with offline reinforcement learning, which likely explains its failure here. In the approximate advantage formulation of DAU, the value and advantage networks are updated with the following loss:
>
> $$Q^i \leftarrow V_{\theta}(s_i) +  \delta t(\bar{A}_\psi(s^i, a^i)-\max_a\bar{A}_\psi(s^i, a))$$
> $$\tilde{Q}^i \leftarrow r^i\delta t + (1-d^i) \gamma^{\delta t} V_\theta (s^{i+1}))$$
> $$ \mathcal{L} = \|| (Q^i-\tilde{Q}^i) \frac 1 {\delta t} ||_2^2 $$
>
> In practice $\tilde{Q}^i$ is a target network updated with Polyak averaging, so the effect of this loss in an offline setting is to minimize $\bar{A}_\psi(s^i, a^i)$ and maximize $\max_a\bar{A}_\psi(s^i, a)$. This directly counteracts the CQL objective, which enforces pessimism on actions that maximize $Q$-values and optimism on actions within the replay buffer. We believe this conflict is why DAU fails. Further research is necessary to come up with a robust version of offline DAU. We’ll include these comparisons in the final version of the paper and discuss the applicability of DAU to the offline RL setting.
>
> > Provide results of exhaustive ablations over the time discretizations and empirical evidence of generalization to discretizations not seen during training.
>
> We would like to clarify that the goal of this work is not necessarily generalization to new frequencies but rather enabling offline RL algorithms to absorb data coming from different frequencies. Nevertheless, this is an interesting question and we ran new experiments to study the generalization to new time discretizations in the kitchen and pendulum environments. We evaluate the learned policies on a range of discretizations, including discretizations outside the range of the training data for FrankaKitchen.
>
> **FrankaKitchen**
> | In Training Data? | $\delta t$ | Adaptive N-Step |
> | ----------- | ----------- | ----------- |
> | No | 45 | 15 $\pm$ 14.8 |
> | Yes | 40 | 34.6 $\pm$ 8.7 |
> | No | 35 | 28 $\pm$ 11.8 |
> | Yes | 30 | 19.9 $\pm$ 7.2 |
> | No | 25 | 2.2 $\pm$ 1.96 |
>
> The policy is capable of interpolating between discretizations seen in training and of extrapolating to coarser, but not necessarily finer grained discretizations. We’ll add these results to the final version of the paper and include the discussion on extrapolation to new discretizations.
>
> We also conduct the same study for all of the frame-skips within the range of training frame-skips on Meta-World Door Open:
>
> **Meta-World Door Open**
> | In Training Data? | $\delta t$ | Adaptive N-Step |
> | ----------- | ----------- | ----------- |
> | Yes | 10 | 89.5 $\pm$ 8.1 |
> | No | 9 | 86.7 $\pm$ 10.3 |
> | No | 8 | 86.7 $\pm$ 10.3 |
> | No | 7 | 100.0 $\pm$ 0.0 |
> | No | 6 | 86.7 $\pm$ 10.3 |
> | Yes | 5 | 92.9 $\pm$ 5.7 |
> | No | 4 | 66.7 $\pm$ 25.8 |
> | No | 3 | 100.0 $\pm$ 0.0 |
> | Yes | 2 | 100.0 $\pm$ 0.0 |
> | Yes | 1 | 87.5 $\pm$ 7.5 |
>
>
> > On a more technical level, the use of N-step returns would make the algorithm on-policy which could break Q-learning based algorithms. It would be helpful if the authors can comment on why this is not the case in their implementation.
>
> This is a great point. While N-step returns technically makes the algorithm on-policy, prior work [1, 2] has found that using N-step returns on off-policy data can work well in practice.
>
> [1] Kalashnikov, D, et al QT-Opt: Scalable Deep Reinforcement Learning for Vision-Based Robotic Manipulation. _Conference on Robot Learning_, 2018.
>
> [2] Espeholt, L, et. al. IMPALA: Scalable Distributed Deep-RL with Importance Weighted Actor-Learner Architectures. _International Conference on Machine Learning_, 2019.

---

> ### Author Response · Authors · 2022-08-25
> **Follow Up to Reviewer Avb9**
>
> Hi reviewer Avb9, We wanted to follow up to see if the response addresses your concerns or if you have any further questions. Thank you again!

---

> > ### Comment · Reviewer_Avb9 · 2022-08-26
> > **Response to Authors**
> >
> > The effort put in by the authors for generating additional experimental results is greatly appreciated. The ablation studies for generalisation to different time discretisations and comparison to a stronger baseline do address a subset of the concerns raised in the review. However, I do concur with Reviewer Z97e that the method is tested only on a few select tasks, and on one of the tasks the proposed methods performs comparable or worse than the naive mixing baseline (depending on the time discretisation), a concern raised in my review as well. This makes the results less convincing. Nonetheless, since the authors have addressed two of the issues, I will change my recommendation to a weak accept.

---

### Official Review · Reviewer_Z97e · 2022-07-30

**Originality:** Good
**Technical Quality:** Good
**Clarity Of Presentation:** Good
**Impact:** 2

**Recommendation:**

Weak Accept: I recommend accepting the paper, but will not argue for my recommendation if the majority of other reviewers have a different opinion.

**Summary:**

This work investigates the problem of offline learning with data collected by different frequencies of control.
The major contributions are:
1. Identifying that directly learning on the mixed data with different control frequencies can downgrade the performance.
2. The authors propose the adaptive N-step update, which adjust the N in N-step TD as well as the discounting.

**Issues:**

Please clarify and address my concerns in the weakness section.

It is said that the authors use door-open from meta-world because it is a straightforward manipulation task. I do not think this is convincing because almost all tasks in meta-world are not considered very hard or long horizon. Personally I find the proposed problem and solution interesting. But the empirical evaluation is the weakest part of this paper at its current form. I think adding experiments on more environments are largely improve its credibility.



**Quality Of The Limitations Section:**

Limitations are addressed clearly

**Reviewer Expertise:**

3: The reviewer is fairly confident that the evaluation is correct

**Robotics Focus:**

Highly relevant to robotics but no hardware experiments

**Strengths And Weaknesses:**

***Strengths***:
1. Offline RL at different frequencies is a relatively overlooked but important area of research. This work confirms the existence of learning challenge under this regime.
2. The proposed method is reasonable, by adapting the N in the N-step TD update, data with different frequencies align better.

***Weakness***:
In general, I think the empirical evaluation is not comprehensive enough to convince me that the proposed problem is universal. Adding more experiments can significantly improve the paper's credibility. Other than this:
1. From the empirical result, it seems the proposed method only works for coarse discretization. So if the data are collected with the same frequency, the proposed method might perform worse than standard offline RL?
2. From my understanding, the proposed method essentially adapts the more on data with more fine-grained discretization. But this usually introduce higher variance as well. I wonder if this is the main problem.
3. As the authors mentioned, the proposed method relies on having a least common multiple for all frequencies. But this assumption is too specific and unlikely holds in practice.

**Summary Of Recommendation:**

I think the proposed problem is overlooked but practical. The proposed solution, though simple, is reasonable from my opinion.

---

> ### Author Response · Authors · 2022-08-23
> **Author Response to Reviewer Z97e**
>
> We greatly appreciate your comments. Based on your suggestions, we have updated our paper to include more details about how our method compares to standard CQL and about the trade-offs of each environment, which has greatly improved the paper clarity.
>
> > It is said that the authors use door-open from meta-world because it is a straightforward manipulation task. I do not think this is convincing because almost all tasks in meta-world are not considered very hard or long horizon. Personally I find the proposed problem and solution interesting.
>
> We agree that Meta-World tasks are not long-horizon. For this reason, the paper also included results on the FrankaKitchen environment with D4RL data (see the rightmost plot of Figure 6 of the main paper). In this environment, there are 4 tasks to solve in sequence: opening a microwave, moving a kettle, turning on a light, and opening a drawer. The average number of environment steps to success is 192.9 in the FrankaKitchen D4RL data and 67.8 in the Meta-World replay data.
>
> > From the empirical result, it seems the proposed method only works for coarse discretization. So if the data are collected with the same frequency, the proposed method might perform worse than standard offline RL?
>
> If the data were all collected with the same frequency, the value of $N$ and all other $\delta t$ based scalings would be identical for each data source, so it would be identical to standard offline RL (and hence also perform the same, not worse). Specifically, in the equations below line 190 we would remove the expectations over the distribution of discretizations $\Delta$. Selecting $N$ to be $\delta t$ in the first equation gives us the standard CQL loss.
>
> > From my understanding, the proposed method essentially adapts the more on data with more fine-grained discretization. But this usually introduce higher variance as well. I wonder if this is the main problem.
>
> This is a great point - your understanding is correct in that the value of $N$ is greater for finer-grained discretizations. This gives a higher variance estimate of the return of the data generating policy for smaller $\delta t$. Appendix figures 1-4 include a Max $N$-Step returns baseline where we use the largest value of $N$ for all discretizations so that the bias-variance tradeoff is identical for each $\delta t$. Adaptive N still outperforms this baseline in terms of speed of convergence in pendulum and final average performance on Kitchen, suggesting that the dependence on $\delta t$ is important. This is especially surprising in the sparse pendulum environment where a larger $N$ should allow the value to propagate along the state space more quickly. We will include this discussion in the final version of the paper and move these baselines to the main paper.
>
> > As the authors mentioned, the proposed method relies on having a least common multiple for all frequencies. But this assumption is too specific and unlikely holds in practice.
>
> When this assumption does not hold, we can sample the next lowest and highest integer with appropriate proportions. We will add these clarifications to the section where we describe this approach on line 191.

---

> ### Author Response · Authors · 2022-08-25
> **Follow Up to Reviewer Z97e**
>
> Hi reviewer Z97e, We wanted to follow up to see if the response addresses your concerns or if you have any further questions. Thank you again!

---

> > ### Comment · Reviewer_Z97e · 2022-08-26
> > **Response to Authors**
> >
> > I appreciate the authors' effort on making additional experiments on the FrankaKitchen domain.
> >
> > To further clarify my concerns:
> >
> > 1. When I mentioned that all tasks in MetaWorld are not considered very hard tasks, I meant that there should be nothing preventing the authors from conducting the same set of experiments on more tasks from MetaWorld. There are 50 tasks in the suite and the authors picked 1 from them, which makes the result less convincing, especially when the 1 result shows only comparable performance of the proposed method versus the naive mixing. Even if the proposed method does not improve much on some of the other tasks, it will be worth mentioning those tasks and provide more analysis on why that happens.
> >
> > 2. Consider the door-open result in Table 1, what confuses me is that the proposed Adaptive N-step method performs worse than naive mixing when $\delta t$ is large (considering the stds, this might be a statistically significant conclusion?). This is why I asked 1 in my original review, please correct me if I have any misunderstanding.
> >
> > In general, I gave weak accept mainly because I think the proposed problem is interesting and could be critical in deployment of RL to real-world and the proposed solution is a reasonable solution to this problem from my opinion. However, as I mentioned in my original review, I think both the writing and the experimental results can be further improved to have a much more solid paper. A good paper should not only propose an interesting/important problem, but should also provide enough evidence if and why the proposed solution to the problem works. If an intuitively correct solution is not working that well as expected, usually digging deep into the failure and providing a detailed analysis can result in a better solution.
> >
> > Therefore I maintain my original score.

---

### Official Review · Reviewer_1JmA · 2022-08-01

**Originality:** Very Good
**Technical Quality:** Good
**Clarity Of Presentation:** Very Good
**Impact:** 4

**Recommendation:**

Weak Accept: I recommend accepting the paper, but will not argue for my recommendation if the majority of other reviewers have a different opinion.

**Summary:**

This paper studies the challenges in RL that come from using offline data collected at different control frequencies. It shows that simplifying running existing algorithms (CQL) on such data (called “naive mixing”) leads to diminished performance. It hypothesizes that the issue is from the inconsistent Q-value targets caused by the mixed control frequencies. They illustrated the hypothesis in a toy example and proposed an adaptive N-step return method to address the issue. The hypothesis and the efficacy of this proposed method are verified in benchmark environments.

**Issues:**

1. Could the authors add results for the original pendulum?
2. Could the authors clarify the state aliasing?


**Quality Of The Limitations Section:**

Limitations are addressed clearly

**Reviewer Expertise:**

3: The reviewer is fairly confident that the evaluation is correct

**Robotics Focus:**

Relevant but unlikely to deploy to hardware in near future

**Strengths And Weaknesses:**

Strengths:
1. The hypothesis is clearly stated and verified in the experiments.
2. The method seems simple and effective.
3. The topic of study (offline RL with data at mixed control freq)  is new to my best knowledge

Weaknesses;
1. It used a modified Pendulum with sparse rewards. It would be nice to see results in the original env especially since it seems from door-open task that dense reward may be “harder” for the method.
2. No theoretical analysis

Minor:
1. I’m not sure if I understood the state aliasing in Fig. 2.
For instance, in the bottom row k=1, why would the unvisited state (second to the left) have the same value as the adjacent state rather than just 0? It’s perhaps a choice for better illustration but I wanted to double-check.
2. Typos
  - line 283: Fig. 6.2 -> 7
  - line 177,178: the top row should have 2 transitions and bottom row should be 1.



**Summary Of Recommendation:**

It is a good paper overall. The problem setting and the hypothesis are clearly stated. The algorithm design and the experiments are nicely set up to verify the hypothesis.
It would make it a stronger paper to include theoretical analysis.

---

> ### Author Response · Authors · 2022-08-23
> **Author Response to Review by Reviewer 1JmA**
>
> Thank you for your feedback. We will update the paper to improve the clarity of the toy examples and explain the motivation behind sparse Pendulum.
>
> > It used a modified Pendulum with sparse rewards. It would be nice to see results in the original env especially since it seems from door-open task that dense reward may be “harder” for the method.
>
> The original Pendulum is an easier task: both Naive Mixing and Adaptive $N$-Step converge in under 10 gradient steps. We modified Pendulum not only to give a more challenging learning environment but also to have a more precise understanding of comparisons between $Q$-values.
>
> > I’m not sure if I understood the state aliasing in Fig. 2. For instance, in the bottom row k=1, why would the unvisited state (second to the left) have the same value as the adjacent state rather than just 0? It’s perhaps a choice for better illustration but I wanted to double-check.
>
> In the grid world example, the agent has the option to take step sizes of up to 2. At the first update step (k=1), the agent can take one step from the second square or two steps from the third square to achieve a reward of 10. These actions are treated as identical, so the reward is discounted in the same way. We will update the description to make this more clear in the final version.

---

> > ### Comment · Reviewer_1JmA · 2022-08-26
> > **Thank you for your clarifications**
> >
> > I thank the authors for the clarifications. My questions were addressed fairly.
> > The concern of the lack of theoretical analysis remains hence I tend to maintain my score as weak accept.
> > A paper with "strong accept" to me would need to have stronger results, which to me means theoretical analysis to some extent (i.e., in the tabular setting).

---

> ### Author Response · Authors · 2022-08-25
> **Follow Up to Reviewer 1JmA**
>
> Hi reviewer 1JmA, We wanted to follow up to see if the response addresses your concerns or if you have any further questions. Thank you again!

---

### Author Response · Authors · 2022-08-23
**Response Summary**

We thank the reviewers for their comments. Based on the reviewer feedback, we’ve added new experiments with:
- Direct comparisons to an offline DAU baseline (Avb9, W1tF)
- Discretization interpolation and extrapolation performance of FrankaKitchen and Meta-World Door policies (Avb9)
- $\delta t$-conditioning for naive mixing and Adaptive $N$-Step on Kitchen (W1tF)
- evaluation of different optimal policies with $\delta t$ conditioning on Kitchen (W1tF)

We will also edit the paper by:
- including discussion of dense Pendulum (1JmA)
- moving Max N-Step returns baseline to the main paper and discussing the bias-variance trade off with larger $N$ (Z97e)

---

### Meta-Review · Area_Chair_3Bep · 2022-08-12

**Recommendation:** Accept (Poster)
**Confidence:** 3

**Metareview:**

AE summarizes the strength and weakness of the paper as follows:

Strength
- The paper is addressing an important and often overlooked topic.
- The paper is well-written. The motivation of the study is clearly presented, and it is easy to understand the proposed method.
- The proposed method seems reasonable

Weakness
- Adjusting N in the N-step return should work for various setting, but the experimental results indicate that the proposed method works only for coarse discretization.
- Experimental results do not include other methods that attempt to be robust against the time discretization.

The paper is addressing the important and often overlooked topic. The paper is overall well-written and easy to follow. For now, the bottleneck is the experimental results. While the experimental results show the advantage of the proposed method to some extent, the paper could be reinforced by adding more experiments.

=== post-rebuttal comments ===

While some concerns were raised by reviewers in the initial review, the authors addressed them by providing the additional experimental results and deepening the discussions. Thus, AE recommends the acceptance of the paper. AE encourages the authors to go through the reviewers comments once again and make sure to reflect the necessary revision in the final manuscript.

**Best Paper Nomination:**

No